# Determinants of Frequent Attendance of Outpatient Physicians: A Longitudinal Analysis Using the German Socio-Economic Panel (GSOEP)

**DOI:** 10.3390/ijerph16091553

**Published:** 2019-05-02

**Authors:** Moritz Hadwiger, Hans-Helmut König, André Hajek

**Affiliations:** 1Department of Health Economics and Health Services Research, Hamburg Center for Health Economics, University Medical Center Hamburg-Eppendorf, Martinistr 52, 20246 Hamburg, Germany; h.koenig@uke.de (H.-H.K.); a.hajek@uke.de (A.H.); 2Institute of Social Medicine and Epidemiology, University of Lübeck, Ratzeburger Allee 160, 23538 Lübeck, Germany

**Keywords:** primary health care, general practitioners, health care utilization, health services needs and demand, primary care, outpatient sector

## Abstract

There is a lack of population-based longitudinal studies which investigates the factors leading to frequent attendance of outpatient physicians. Thus, the purpose of this study was to analyze the determinants of frequent attendance using a longitudinal approach. The used dataset comprises seven waves (2002 to 2014; *n* = 28,574 observations; ranging from 17 to 102 years) from the nationally representative German Socio-Economic Panel (GSOEP). The number of outpatient physician visits in the last three months was used to construct the dependent variable “frequent attendance”. Different cut-offs were used (top 25%; top 10%; top 5%). Variable selection was based on the “behavioral model of health care use” by Andersen. Accordingly, variables were grouped into predisposing, enabling, and need characteristics as well as health behavior, which are possible determinants of frequent attendance. Conditional fixed effects logistic regressions were used. As for predisposing characteristics, regressions showed that getting married and losing one’s job increased the likelihood of frequent attendance. Furthermore, age was negatively associated with the outcome measure. Enabling characteristics were not significantly associated with the outcome measure, except for the onset of the “practice fee”. Decreases in mental and physical health were associated with an increased likelihood of frequent attendance. Findings were robust across different subpopulations. The findings of this study showed that need characteristics are particularly important for the onset of frequent attendance. This might indicate that people begin to use health services frequently when medically indicated.

## 1. Introduction

A large share of outpatient physician consultations can be attributed to a small share of patients, resulting in situations where a lot of resources are concentrated on a few patients. Patients responsible for this behavior are called frequent attenders [1,2,3,4,5].

Determinants of frequent attendance can be derived theoretically from the “behavioral model of health service use” by Andersen et al. (2014) [6]. The model is commonly used for the analysis of health utilization behavior [7]. Variables for analyzing health care use are grouped into predisposing (e.g., age, sex), enabling (e.g., income, insurance status), and need characteristics (e.g., morbidity, self-rated health) as well as health behavior. These three different types of factors and health behaviors serve as the possible determinants for analyzing frequent attendance.

With respect to need characteristics, frequent attenders are described as patients whose burden of physical and psychological morbidity is high compared to the non-frequent attenders. They are often characterized by multimorbidity, depression, anxiety, or severe chronic illnesses. In the context of frequent attendance, self-rated health is crucial in order to understand the use patterns of frequent attendance [1,8,9,10]. As for predisposing characteristics, the results are more ambiguous. Findings differ with respect to sex, age, and marital status, but several studies summarized in a review suggest that frequent attenders are more likely to be female [1].

In sum, the phenomenon of frequent attendance is widely researched but not comprehensively understood [3]. More specifically, most publications in the field of frequent attendance are restricted to a cross-sectional design [1,11]. There is a lack of studies analyzing the phenomenon of frequent attendance using longitudinal data [8,12,13]. Using longitudinal data is important to tackle the issue of unobserved heterogeneity (e.g., genetic disposition) and to gain insights into intraindividual changes over time (e.g., changes within individuals over time from non-frequent attendance to frequent attendance). Beyond that, a lot of studies in this field suffer from small sample sizes [14]. Consequently, the aim of this study was to identify the determinants of frequent attendance longitudinally based on a population-based sample. Knowing the factors that lead to frequent attendance may help to reduce the economic burden associated with frequent attendance. When frequent attendance is, for example, driven by predisposing and enabling characteristics, this might point to over- or misuse of health care services. When frequent attendance is driven by modifiable need factors, postponing these need-factors might be beneficial for the health care system.

In Germany health insurance is mandatory. Nearly 90% of the population is insured by statutory health insurance (SHI) institutions. The remaining 10% of the population is insured by private health insurance (PHI). Access to the whole health care system is provided for all insured individuals. The contribution for statutory insurance is accounted for by payroll tax and is independent from the risk of the individual. Non-employed family members are equally insured without additional premiums. There is no barrier, such as gatekeeping, in Germany for patients, but older patients are especially likely to use referrals by primary care physicians for a specialist visit. The ambulatory health care is a profit-based service provided by private physicians. The density of physicians in Germany is considerably above the average compared to other member states of the European Union. In Germany, outpatient physician visits are high when compared internationally. Between the years 2004 and 2012, patients had to pay 10 Euro once a quarter for a physician visit and an additional 10 Euro for a specialist visit without referral. Additionally, small copayments are charged for prescribed drugs [15,16].

## 2. Materials and Methods

### 2.1. Sample

The data collection of the population-based German Socio-Economic Panel (GSOEP) started in 1984 by the German Institute for Economic Research (DIW Berlin). It is a longitudinal, representative, and annually conducted survey for private household data [17]. Individuals aged 17 and over were included. Panel attrition in the GSOEP is low, even if the health status of the individual worsens [18]. The GSOEP contains an array of socio-demographic variables. The micro data at the household and individual level provide the possibility to analyze economic and social behavior.

The variables for the SF-12v2 (12-Item Short Form Survey, version two) only became available in even years since 2002 [18]. Due to this, the following analysis was restricted to 2002 until 2014 (i.e., 2002, 2004, 2006, 2008, 2010, 2012, and 2014; *n* = 28,574 observations).

Participants gave their informed consent prior to data collection. Compliant with national laws as well as evaluated and approved by the German Council of Science and Humanities (Wissenschaftsrat), the GSOEP is ethically sound and explicitly intended for epidemiological analyses. All procedures contributing to this work adhered to the Declaration of Helsinki.

### 2.2. Dependent Variable

To construct the dichotomous variable “frequent attender” (no/yes), the following question for outpatient utilization in the GSOEP was used:

“Have you consulted a physician within the last three months? If yes, please state how often.”

Six or more physician consultations over the whole observation horizon represent the 90th percentile. To identify a frequent attender, the 90th percentile cut-off threshold for physician consultations was selected, because it was widely used in previous studies [4]. In order to check the robustness of our results, thresholds for nine or more visits (approx. 95th percentile) and for three or more visits (approx. top 75th percentile) were used as well.

### 2.3. Independent Variables

The independent variables for the analysis of frequent attendance were derived from the “behavioral model of health service use” by Andersen et al. (2014) [6]. The selection was also guided by the literature on frequent attendance [1,7]. Predisposing, enabling, and need characteristics as well as health behavior served as the possible determinants of frequent attendance.

#### 2.3.1. Predisposing Characteristics

Age, sex, and migration background were included [2,6]. Although the last two variables are time-invariant (i.e., they do not vary within individuals over time), they were reported in the descriptive results. Moreover, marital status was included. To this end, the variable “partner” (no/yes) was used (married or living together with a partner; other (single, separated, divorced, widowed)). To capture possible effects of unemployment [1,19], a variable for non-working (no/yes) was included. The individuals can have the status of non-working, for example, due to unemployment or retirement with reference to all kinds of employment. Furthermore, a variable “university” (no; yes) for tertiary education was included [8].

#### 2.3.2. Enabling Characteristics

The net equivalent disposable income per household member was used. It was derived from the household net income in Euro divided by the square root of the household size. In regression analysis, the logarithm of the net equivalent disposable income was used, because of diminishing marginal returns of a rising income. The idea of the net equivalent income is that individuals who share a household have a kind of economics of scale [20,21]. Private health insurance (PHI) (ref. statutory insured) was included to capture effects such as supplier-induced demand [22]. Dummies for the beginning of the “practice fee” in 2004 and the end of the “practice fee” in 2012 were used (period effects) in order to capture potential changes in the utilization behavior [16,23].

#### 2.3.3. Need Characteristics

For evaluated need characteristics, severe disability (no/yes) was used as a surrogate for morbidity [24,25,26,27].Being disabled is covered in the GSOEP by asking participants whether they are legally classified as being a disabled person.

For perceived need, the two subordinate scales of the SF-12v2 were used, “Mental Health Composite Score” (MCS) and “Physical Health Composite Score” (PCS). Single items and subscales were computed to the two main dimensions using norm-based scoring and factor analysis. Values of the two scales have a range between 0 and 100 with a mean of 50 and a standard deviation of 10 [18,28]. To ease the interpretation, reversed scales were used (MCS reversed ranging from 1.27 to 79.63, with higher values corresponding to worse mental health; PCS reversed ranging from 9.20 to 79.60, with higher values corresponding to worse physical health), because in the original scale an increase of one of the scales would indicate a worse health status. The SF-12 is referred as a “quasi-objective” health measure [29].

#### 2.3.4. Health Behavior

To cover negative health behavior, the variable non-smoking (no/yes) observed if an individual stopped smoking during the time horizon of the analysis. The body mass index (BMI) was calculated by the answers of the individuals about their weight (kg) and height (meters) (weight divided by height-squared). For a better interpretation, BMI was centered.

### 2.4. Statistical Analysis

Panel data offer the opportunity to use the within estimator (which is also known as the fixed effects (FE) estimator). This estimator is not distorted by time-invariant, person-specific unobserved heterogeneity. The within estimator estimates causal effects (with certain restrictions) by intraindividual comparisons [30]. This means that the results of the FE regressions can be interpreted as average treatment effects on the treated [23]. The within estimator uses only the variation over time and all time-invariant components are removed [31]. Some examples are time-constant unobserved variables, such as genetic disposition, which can drive health care utilization [6,31]. Not considering unobservable time-invariant variables like genetic disposition can lead to biased estimates [20].

It is crucial for the within estimator that the explanatory variables used in the model have sufficient variation across time. Little within-variation of the independent variables can lead to biased estimates [32].

Due to the dichotomous characteristic of the dependent variable, a logit regression model with person-specific errors was applied. A logit regression estimates the effect of several independent variables on the unobserved probability of the binary dependent variable [31]. By using a conditional likelihood function, the person-specific errors cancel out. Thus, the estimator is consistent, even if it is correlated with time-invariant, person-specific unobserved heterogeneity [33]. It should be noted that individuals who have the same constant dependent result over the whole observation period are excluded from the estimation. This can reduce the dataset but does not evoke sample selection bias, because individuals do not contain information for the estimation of the coefficients [31].

Time-constant variables, such as sex or migration status, cannot be included as main effects in FE regressions (because they did not vary within individuals over time), although interaction terms between time-variant and time-invariant variables can be included in FE regressions [33]. The significance level was set at *p* < 0.05. Stata/SE 14.2 (Stata Corp., College Station, TX, USA) was used for the estimation.

## 3. Results

### 3.1. Descriptive Statistics

Due to the fact that FE regression analysis of frequent attendance only includes individuals who have a variation in the dependent variable over time, the descriptive results in Table 1 display a subsample of the GSOEP. The descriptive results (pooled) stem from the main analysis of the threshold definition of frequent attendance (six or more quarterly physician visits). The analysis includes all even years between 2002 to 2014 (seven waves). There were *n* = 28,574 observations (6179 individuals).

The average frequency of physician visits in the last three months was 4.6 (±5.5) and the median was three. As regards the predisposing characteristics, the majority of the sample was female (55.6%), and the mean age was 54 years (±16.7) with a range from 17 to 102 years. Most of the individuals were married or living together with a partner, and a slight majority was non-working (51.1%). Additionally, 20.5% of the individuals had a university degree and 17.3% had a migration background. As for enabling characteristics, the mean net equivalent disposable income per household member was 1795.50 Euro (SD ±1373.00 Euro), with a median of 1500 Euro per month. In the sample, 14% of the individuals have a private health insurance.

As for need characteristics, the mean reversed score for MCS was 32.2 (±11.1) and for PCS the mean reversed score was 44.05 (±10.88). The mean BMI score was 26.6 (±5.0) and the median was 26.0.

### 3.2. Regression Analysis

The variable for a university degree has strikingly little within variation. Hence, it was not included in the FE regression. Multicollinearity was not detected, according to low bivariate correlation values of the included variables and below common threshold values of the variance inflation factor [20,34]. The results of the Hausman test (Stata command: hausman) strongly reject the null hypothesis that a random effects (RE) model should be applied (*p* < 0.001). This result suggests the use of a FE estimator [35].

Table 2 displays the regression results. The pseudo-*R*^2^ varied between 0.102 and 0.162. The first model (1) reports the results for the frequent attender definition with six or more visits, which corresponds approximately to the 90th percentile. To check for different threshold definitions, the other two models (2) (approx. 95th percentile) and (3) (approx. 75th percentile) were also reported.

The regression results are displayed in odds ratios (ORs). For model (1), there were 28,574 observations for 6179 individuals. For model (2), there were 16,496 observations for 3550 individuals, and for model (3) the number of observations was 57,323 for 12,542 individuals. The results are reported for the main model (1). If there were deviations in the results of the other two models, they are reported as well.

As for predisposing characteristics, an increase in age decreased the odds of being a frequent attender by 0.949 (*p* < 0.001), holding all other variables constant. Getting married (or getting a partner) increased the odds for becoming a frequent attender by 1.228 (*p* < 0.001). A shift from working to non-working increased the odds by 1.352 (*p* < 0.001).

With respect to enabling characteristics, neither changes in income nor changes in insurance status were significantly associated with the outcome measure. The OR of the period effect for the year 2004 was 0.889 (*p* < 0.01), which indicates a decline in the likelihood of becoming a frequent attender in the year 2004. The dummy for the year 2012 was not significant.

As for need characteristics, a higher score of the reversed scale in MCS and PCS was associated with a higher probability of being a frequent attender. For a one unit increase on the reversed scale, the odds for being a frequent attender increased by a factor of 1.049 (*p* < 0.001) and 1.117 (*p* < 0.001) for MCS and PCS, respectively. Becoming legally classified as severely disabled increased the odds by 1.518 (*p* < 0.001). However, this association was only significant in model (3). With regard to health behavior, quitting smoking increased the odds for becoming a frequent attender by the factor of 1.343 (*p* < 0.001).

The robustness of the results was tested by comparing the model ((1); see Table 2) with additional models for different subpopulations for the same threshold definition of frequent attendance (90th percentile). For time-invariant variables, regressions were conducted separately for males, females, and individuals with migration background: regressions showed that results remained virtually the same.

When testing for attrition bias, the regression analysis was only carried out for individuals whose dependent variable is observed for the whole observation period. The results do not differ substantially from the main regression. Getting married or having a long-term partner became insignificant.

In the last step, interaction-terms for sex and MCS and PCS were added to the main model. However, the interaction terms did not reach statistical significance.

## 4. Discussion

Using a large population-based longitudinal dataset (GSOEP), this study aimed to identify the determinants of frequent attendance using panel data methods. By doing that, this study provides additional insights on how the different determinants (predisposing, enabling, and need characteristics as well as health behavior) affected frequent attendance.

### 4.1. Predisposing Characteristics

Concerning predisposing factors, the results were mixed. Increasing age led to a decline in the likelihood of frequent physician visits in our study. This result contrasts with the literature. Generally, older individuals are linked to frequent attendance [5,9]. However, Hajek et al. (2017) [26] also found that with increasing age, physician visits decrease for the GSOEP. A possible explanation could be that with increasing age individuals got more frustrated with the health care system in Germany. Another explanation might be that the access to outpatient physicians might decrease with age.

Getting married and changes to non-working were both associated with an increased likelihood of frequent attendance in our study. The link between employment status and frequent attendance builds upon previous studies that were often restricted to a cross-sectional design [1,8,14]. A possible explanation for the increased likelihood of becoming a frequent attender post marriage could be an increased awareness regarding one’s personal health status due to the additional health-related advise provided by the spouse.

### 4.2. Enabling Characteristics

Except for the period effect of the introduction of the “practice fee” [10], enabling characteristics were not associated with the outcome measure in our study. This can be explained by the fact that individuals insured with the statutory health insurance can easily consult a physician free of public copayment.

### 4.3. Need Characteristics

The main model (1) showed that perceived need factors are an important determinant in the context of frequent attendance in our study. Quasi-objective generic health measures are crucial for understanding the patterns of frequent attendance. Decreases in mental and physical health were associated with an increased likelihood of frequent attendance. These results add to previous knowledge, which has mainly been based on cross-sectional studies [36]. This was also supported by two longitudinal studies. These studies found that a decline of the SF-12 score was associated with frequent attendance [8,13].

### 4.4. Health Behavior

The conclusions for determinants of health behavior for the probability of becoming a frequent attender are limited. Stopping smoking was associated with an increased likelihood of becoming a frequent attender. One possible explanation could be that the individual stopped smoking because of health problems (analogous to the “sick quitter effect”, which has often been observed among non-drinkers) and subsequently had to visit physicians frequently. However, information is missing about how long, or how much the individuals smoked, or why he or she stopped smoking. Future research is required to clarify this issue.

### 4.5. Strengths and Limitations

A FE regression approach was used in this study. This was done to control for unobserved as well as for observed time-invariant person-specific variables, such as genetic disposition. These factors are expected to bias the estimates [35]. This is the main strength of the analytical method chosen [31]. A large population-based sample was used. It was also checked for panel attrition via a sensitivity analysis. Furthermore, different cut-offs were used for frequent attendance. Moreover, the validated SF12 was used to quantify mental and physical health.

However, a potential bias due to reverse causality of the estimates cannot be ruled out. For example, consulting a physician frequently could eventually lead to a decrease in physical health. If there is a source of endogeneity in this analysis, an instrumental variable approach can be applied, but this method relies on strong assumptions. When these strong assumptions are not fulfilled, results are heavily biased [23].

Despite the rich set of sociodemographic variables included in this analysis, the analysis lacks evaluated need factors, such as chronic diseases, due to data unavailability. A further limitation is that there is no differentiation in the dataset between primary physicians and specialists. Regarding the dependent variable, only a three-month period for physician visits was captured. However, a three-month time horizon was often used in previous studies. Furthermore, it has been demonstrated that the time horizon is suitable and the recall bias is negligible [37]. Moreover, future studies are required to clarify the determinants of frequent hospital visits longitudinally.

## 5. Conclusions

The findings of this study showed that need characteristics are particularly important for the onset of frequent attendance. This might indicate that people may be especially likely to begin to use health services frequently when medically indicated. Consequently, finding ways to increase health may help to reduce the economic burden caused by frequent attendance

Researching determinants for frequent attendance is the first step towards understanding frequent attendance. Future studies are required to investigate the determinants of persistent frequent attenders.

## Figures and Tables

**Table 1 ijerph-16-01553-t001:** Descriptive results (2002–2014 GSOEP).

	Variables	Mean	Std. Dev.	Min	Max
	Physician Visits ^1^	4.648	5.508	0	99
Predisposing characteristics	Sex (1 = female)	0.556	0.497	0	1
	Age (in years)	53.58	16.72	17	102
	Partner (1 = yes) ^2^	0.637	0.481	0	1
	Non-working (1 = yes) ^3^	0.511	0.500	0	1
	University (1 = yes) ^4^	0.205	0.404	0	1
	Migration (1 = yes) ^5^	0.173	0.378	0	1
Enabling characteristics	Equivalence income (in €)	1795.47	1372.95	75	70,709.97
	PHI ^6^ (1 = yes) (ref. SHI) ^7^	0.140	0.347	0	1
Need characteristics	MCS reversed ^8^	32.15	11.13	1.467	77.01
	PCS reversed ^9^	44.05	10.88	13.53	78.43
	Disabled (1 = yes)	0.224	0.417	0	1
Health behavior	Smoking (1 = yes)	0.318	0.466	0	1
	BMI	26.60	5.017	12.41	136.8
Observations	28,574				

Notes: ^1^ During the last three months. ^2^ Ref. single, widowed, divorced, separated. ^3^ Ref. full-time working, part-time working, apprenticeship, marginal employment, sheltered workshop. ^4^ The variable “university” has 28,570 observations. ^5^ The variable “migration” has 28,570 observations. ^6^ PHI: Private Health Insurance. ^7^ SHI: Statutory Health Insurance. ^8^ Mental Health Composite Score. ^9^ Physical Health Composite Score. GSOEP: German Socio-Economic Panel.

**Table 2 ijerph-16-01553-t002:** Regression results GSOEP 2002–2014: frequent attenders.

	(1)	(2)	(3)
Variables	Frequent Attender 90th percentile ^11^	Frequent Attender 95th percentile ^12^	Frequent Attender 75th percentile ^13^
Age (in years)	0.949 *** (0.939–0.958)	0.931 *** (0.918–0.944)	0.984 *** (0.978–0.991)
Partner (1 = yes) ^14^	1.228 ** (1.066–1.414)	1.217 * (1.005–1.474)	1.003 (0.913–1.101)
Non-working (1 = yes) ^15^	1.352 *** (1.223–1.495)	1.350 *** (1.179–1.545)	1.266 *** (1.186–1.351)
Log equivalence income (in €)	0.996 (0.914–1.085)	0.979 (0.870–1.102)	1.025 (0.970–1.083)
PHI ^16^ (1 = yes) (ref. SHI ^17^)	1.011 (0.766–1.335)	1.024 (0.690–1.521)	0.865+ (0.738–1.014)
MCS reversed ^18^	1.049 *** (1.045–1.052)	1.051 *** (1.046–1.057)	1.041 *** (1.038–1.043)
PCS reversed ^19^	1.117 *** (1.112–1.123)	1.122 *** (1.115–1.129)	1.103 *** (1.099–1.106)
Disabled (1 = yes)	1.052 (0.921–1.200)	0.985 (0.833–1.165)	1.518 *** (1.355–1.701)
Non-smoking (1 = yes)	1.343 *** (1.176–1.534)	1.308 ** (1.095–1.563)	1.353 *** (1.243–1.472)
BMI centered	0.991 (0.976–1.006)	0.981 * (0.962–1.000)	0.995 (0.985–1.006)
Year 2004 ^20^	0.889 ** (0.822–0.960)	0.894 * (0.806–0.991)	0.909 *** (0.864–0.958)
Year 2012 ^21^	1.018 (0.914–1.133)	0.990 (0.855–1.147)	1.019 (0.948–1.094)
Observations	28,574	16,496	57,323
Number of individuals	6179	3550	12,542
Pseudo-*R*^2^	0.145	0.162	0.102

Odds ratios were reported; 95% Confidence intervals in parentheses; *** *p* < 0.001, ** *p* < 0.01, * *p* < 0.05, + *p* < 0.10, Hausman test: (1): 244.42*** (2): 214.12*** (3): 236.30*** Notes: ^11^ Frequent attender (absolute threshold definition: six or more visits approx. 90th percentile). ^12^ Frequent attender (absolute threshold definition: nine or more visits approx. 95th percentile). ^13^ Frequent attender (absolute threshold definition: three or more visits approx. 75th percentile). ^14^ Ref. single, widowed, divorced, separated. ^15^ Ref. full-time working, part-time working, apprenticeship, marginal employment, sheltered workshop. ^16^ PHI: Private Health Insurance. ^17^ SHI: Statutory Health Insurance. ^18^ Mental Health Composite Score. ^19^ Physical Health Composite Score. ^20^ Ref. 2002, 2006, 2008, 2010, 2012, 2014. ^21^ Ref. 2002, 2004, 2006, 2008, 2010, 2012.

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
