# Peer review of "Determinants of Frequent Attendance of Outpatient Physicians: A Longitudinal Analysis Using the German Socio-Economic Panel (GSOEP)"

_ijerph, 2019, doi:10.3390/ijerph16091553_

Round 1
Reviewer 1 Report
This was an interesting study to review and well designed. I found the use of services declined as one aged to be interesting and I think this would warrant further research.. Lines 240-241 "additional reflections" of the spouse is not clear. Would you strengthen this statement so if is not ambiguous? I feel the conclusion could be stronger by emphasizing the "so what" factor of the study's findings.
Author Response
RESPONSES TO COMMENTS OF REVIEWERS
Comments of reviewers | Response (citations from manuscript printed in italics, changes are underlined) |
Reviewer #1 | |
This was an interesting study to review and well designed. I found the use of services declined as one aged to be interesting and I think this would warrant further research. | First, thank you very much to review this manuscript. We really appreciate your comments. |
Lines 240-241 "additional reflections" of the spouse is not clear. Would you strengthen this statement so if is not ambiguous? | Thank you for your comment. We changed it accordingly: A possible explanation for the increased likelihood of becoming a frequent attender post marriage could be an increased awareness regarding the personal health status due to the additional health related advises provided by the spouse |
I feel the conclusion could be stronger by emphasizing the "so what" factor of the study's findings. | Thank you for your comment. Findings of this study showed that need characteristics are particularly important for the onset of frequent attendance. This might indicate that persons begin to use health services frequently especially when medically indicated. Consequently, finding ways to increase health may help to reduce the economic burden caused by frequent attendance Again, thank you for your helpful comments. It helps to improve the quality of the manuscript. |

Reviewer 2 Report
This study investigates determinants of frequent attendance of outpatient physicians based on data from the German Socio-Economic Panel. Findings suggest that frequent attendance is particularly associated with subjectively perceived health status.
The study is relevant and straight forward with sound methods. I only have a few comments the authors should address to improve the manuscript.
I wish the authors could give us some more information on how the SF-12 physical and mental component summary scales were calculated. Moreover, it seems that in the reversed scales higher scores would indicated worse and not better health.
It would also be good if the author could explain the statistical model in more detail and could describe how they tested for multi-collinearity as well as the application of the Hausman test.
It would make the tables more readable if the authors could use superscript for the footnotes.
I also do not understand why in the discussion as well as in the abstract it is said that enabling characteristics were not significantly associated with the outcome. There is obviously a period effect for the year 2004 indicating that the introduction of the practice fee might have reduced attendance of outpatient physicians.
A further limitation that the authors could discuss is that hospitalizations were not considered in this study.
Author Response
RESPONSES TO COMMENTS OF REVIEWERS
Comments of reviewers | Response (citations from manuscript printed in italics, changes are underlined) |
Reviewer #2 | |
I wish the authors could give us some more information on how the SF-12 physical and mental component summary scales were calculated. | Thank you for your comment. We added: Single items and subscales are computed to the two main dimensions using norm-based scoring and factor analysis. Values of the two scales have a range between 0 and 100 with a mean of 50 and a standard deviation of 10 [18,28]. |
Moreover, it seems that in the reversed scales higher scores would indicated worse and not better health. | Thank you for your comment. We changed it accordingly: reversed scales were used (MCS reversed ranging from 1.27 to 79.63, with higher values corresponding to worse mental health; PCS reversed ranging from 9.20 to 79.60, with higher values corresponding to worse physical health) |
It would also be good if the author could explain the statistical model in more detail | Thank you for your comment. We added: Due to the dichotomous characteristic of the dependent variable a logit regression model with person specific errors was applied. A logit regression estimates the effect of several independent variables on the unobserved probability of the binary dependent variable [31]. Using a conditional likelihood function the person specific errors cancels out. Thus, the estimator is consistent, even if it is correlated with time-invariant, person-specific unobserved heterogeneity [33]. It should be noted that individuals who have the same constant dependent result over the whole observation period are excluded from the estimation. This can reduce the dataset but does not evoke sample selection bias, because individuals do not contain information for the estimation of the coefficients [31]. |
how they tested for multi-collinearity as well as the application of the Hausman test | Thank you for your comment. We added: Multicollinearity was not detected, according to low bivariate correlation values of the included variables and below common threshold values of the Variance Inflation Factor The results of the Hausman-Test (Stata command: hausman) strongly reject the null hypothesis that a random effects (RE) model should be applied (p<0.001). This result suggests the use of a FE estimator [35]. |
It would make the tables more readable if the authors could use superscript for the footnotes. | Thank you for your comment. We changed it accordingly. Please see the Tables for further details.
|
I also do not understand why in the discussion as well as in the abstract it is said that enabling characteristics were not significantly associated with the outcome. There is obviously a period effect for the year 2004 indicating that the introduction of the practice fee might have reduced attendance of outpatient physicians. | Thank you for your comment. We rearranged the sentence: Except for the period effect of the introduction of the “practice fee” [10], enabling characteristics were not associated with the outcome measure in our study. That can be explained by the fact that individuals insured with the statutory health insurance can easily consult a physician free of public copayment. And added a subordinate clause in the abstract: Enabling characteristics were not significantly associated with the outcome measure, expect for the onset of the “practice fee”. |
A further limitation that the authors could discuss is that hospitalizations were not considered in this study. | Thank you for your comment. We added: Despite the rich set of sociodemographic variables included in this analysis, the analysis lacks evaluated need-factors, such as chronic diseases due to data unavailability. A further limitation is that there is no differentiation in the dataset between primary physicians and specialists. Regarding the dependent variable, only a three-month period for physician visits was captured. However, a three month time horizon was often used in previous studies. Furthermore, it has been demonstrated that the time horizon is suitable and the recall bias is negligible [37]. Moreover, future studies are required to clarify the determinants of frequent hospital visits longitudinally. Again, thank you for your helpful comments. It helps to improve the quality of the manuscript. |
